# MOGAT: A Multi-Omics Integration Framework Using Graph Attention Networks for Cancer Subtype Prediction

**DOI:** 10.3390/ijms25052788

**Published:** 2024-02-28

**Authors:** Raihanul Bari Tanvir, Md Mezbahul Islam, Masrur Sobhan, Dongsheng Luo, Ananda Mohan Mondal

**Affiliations:** Knight Foundation School of Computing and Information Sciences, Florida International University, Miami, FL 33199, USA; rtanv003@fiu.edu (R.B.T.); misla093@fiu.edu (M.M.I.); msobh002@fiu.edu (M.S.)

**Keywords:** cancer subtype prediction, graph neural network, graph attention network, multi-omics integration

## Abstract

Accurate cancer subtype prediction is crucial for personalized medicine. Integrating multi-omics data represents a viable approach to comprehending the intricate pathophysiology of complex diseases like cancer. Conventional machine learning techniques are not ideal for analyzing the complex interrelationships among different categories of omics data. Numerous models have been suggested using graph-based learning to uncover veiled representations and network formations unique to distinct types of omics data to heighten predictions regarding cancers and characterize patients’ profiles, amongst other applications aimed at improving disease management in medical research. The existing graph-based state-of-the-art multi-omics integration approaches for cancer subtype prediction, MOGONET, and SUPREME, use a graph convolutional network (GCN), which fails to consider the level of importance of neighboring nodes on a particular node. To address this gap, we hypothesize that paying attention to each neighbor or providing appropriate weights to neighbors based on their importance might improve the cancer subtype prediction. The natural choice to determine the importance of each neighbor of a node in a graph is to explore the graph attention network (GAT). Here, we propose MOGAT, a novel multi-omics integration approach, leveraging GAT models that incorporate graph-based learning with an attention mechanism. MOGAT utilizes a multi-head attention mechanism to extract appropriate information for a specific sample by assigning unique attention coefficients to neighboring samples. Based on our knowledge, our group is the first to explore GAT in multi-omics integration for cancer subtype prediction. To evaluate the performance of MOGAT in predicting cancer subtypes, we explored two sets of breast cancer data from TCGA and METABRIC. Our proposed approach, MOGAT, outperforms MOGONET by 32% to 46% and SUPREME by 2% to 16% in cancer subtype prediction in different scenarios, supporting our hypothesis. Our results also showed that GAT embeddings provide a better prognosis in differentiating the high-risk group from the low-risk group than raw features.

## 1. Introduction

Integrating multi-omics data is crucial for gaining a comprehensive understanding of complex diseases like Alzheimer’s [1], Parkinson’s [2,3], and cancer [4]. However, it is a difficult task that requires advanced computational methods. New analytical tools and methods are needed to effectively extract biologically relevant information from multi-omics data and integrate it into a comprehensive understanding of the disease. Despite the challenges of the high dimensionality and complexity of data, the integration of multi-omics data holds great potential for understanding the biology of cancer.

Graph-based learning models are used in many proposed models to get hidden representations and graph structures from different omics data. This helps us learn more about Alzheimer’s, Parkinson’s, cancer prediction, patient categorization, and other topics. Wang et al. [1] used multi-omics integration for Alzheimer’s disease patient classification. Researchers also used multi-omics integration to find molecular biomarkers [3] and disease classification [2] for Parkinson’s disease. Much multi-omics research has been conducted to predict cancer subtypes and patient categorization. Li et al. [5] utilized a graph convolutional network (GCN) [6] to classify 28 different cancer types from pan-cancer data using gene expression and copy number alteration as features and three knowledge networks as the input graphs, including gene–gene interaction (GGI) networks, protein–protein interaction (PPI) networks, and gene co-expression networks. Zhou et al. [7] used gene expression, DNA methylation, and miRNA expression as features for multi-omics analysis. They used anchors to derive sample similarity networks and a graph convolutional autoencoder for clustering cancer samples to identify novel subtypes for breast, brain, colon, and kidney cancer. Guo et al. [8] used GCN by taking the PPI network as a graph and gene expression, copy number alterations, and DNA methylation as features. Finally, they applied attention on top of embeddings generated by GCN to classify breast cancer subtypes. Li et al. proposed the MoGCN [9], which uses an autoencoder for feature extraction and similarity network fusion to construct the patient similarity network. It applies GCN to classify breast cancer subtypes and pan-kidney cancer type classification using gene expression, copy number alterations, and phase protein array data as input. M-GCN [10] is another multi-omics framework based on GCN to classify breast and stomach cancer subtypes. They use the Hilbert-Schmidt independence criterion-based least absolute shrinkage and selection operator (HSIC LASSO) to select the molecular subtype-related transcriptomic features and then use those features to construct a patient similarity graph applying Pearson’s correlation. It takes gene expression, single nucleotide variation, and copy number alterations as input for multi-omics data. MOGONET [1] inputs gene expression, DNA methylation, and miRNA. Unlike other methods, it uses GCN to learn omics-specific embeddings and uses network and node features for particular omics data. Then, it combines the embeddings using a view correlation discovery network (VCDN) to classify cancer subtypes for breast, brain, and pan-kidney cancer. The SUPREME [11] method utilizes GCN for analyzing breast cancer subtypes. It integrates seven types of data—gene expression, miRNA expression, DNA methylation, single nucleotide variation, copy number alteration, co-expression module eigengenes, and clinical data—for constructing the network and determining node features. Additionally, it combines GCN embeddings with node features and employs a multi-layer perceptron (MLP) as a classifier.

In summary, the existing GNN-based multi-omics integration approaches to predict cancer subtypes apply GCN to extract salient features from different omics data. However, GCN-based frameworks cannot determine the relative significance of neighboring samples regarding downstream analyses, including cancer subtype prediction, patient stratification, etc. It is also noticeable that none of the existing studies considered long non-coding RNA (lncRNA) expression data in multi-omics integration. However, lncRNAs play important regulatory roles in various cellular processes, including gene expression and epigenetic regulation [12,13,14,15].

This research presents MOGAT, illustrated in Figure 1, a novel multi-omics integration-based cancer subtype prediction leveraging a graph attention network (GAT) [16] model that incorporates graph-based learning with an attention mechanism for analyzing multi-omics data. The proposed MOGAT utilizes a multi-head attention mechanism that can extract information for a specific patient more efficiently by assigning unique attention coefficients to its neighboring patients, i.e., obtaining the relative influence of neighboring patients in the patient similarity graph. We also include lncRNA expression in the multi-omics integration process. Altogether, eight different data types are integrated, including mRNA expression, miRNA expression, lncRNA expression, DNA methylation, single nucleotide variation, copy number alteration, co-expression module eigengenes, and clinical data. Based on our knowledge, only one other multi-omics integration framework utilizes GAT to identify cancer driver genes but not for cancer subtype prediction [17].

The salient features of this study are enumerated below.

Our group is the first to explore graph attention network-based multi-omics integration for cancer subtype prediction.The proposed approach, MOGAT, provides better embeddings than MOGONET and SUPREME for multi-omics integration, which results in improved accuracy for cancer subtype prediction.MOGAT embeddings provide a better prognosis in differentiating the high-risk group from the low-risk group, which will help the physician devise an appropriate treatment strategy for an individual patient depending on the location of the patient on the prognostic curve.Our group is the first to incorporate lncRNA expression in multi-omics integration studies.We provided detailed information so that the results can be reproduced, such as (a) handling duplicate samples coming from the same patient and (b) providing the number of features in each step of preprocessing, from raw features to cleaned features to selected features.The interactions between different omics types are considered during the node feature engineering by concatenating features from different omics types.

## 2. Results

### 2.1. Comparison of Performance

To assess the performance of the proposed MOGAT framework, we compared it with two state-of-the-art frameworks that integrate multi-omics data for cancer subtype prediction, namely, MOGONET and SUPREME. The macro-F1 score is used to compare the performance, as illustrated in Table 1 with average (avg) and standard deviation (SD), as well as in Figure 2 with violin plots for different combinations of omics data. Three omics data were used to show the performance comparison of MOGAT with MOGONET and SUPREME: gene expression, DNA methylation, and miRNA expression, as they were originally used in MOGONET. For the same three omics data, the test macro-F1 scores on seven ($2^{3}$ − 1) combinations of three omics data were calculated and plotted in Figure 2a. For 3-omics analysis, we observed that (Table 1) MOGAT has higher macro-F1 scores with an average of 0.804 compared to 0.550 and 0.732 for MOGONET and SUPREME, respectively.

However, for all the omics data, a comparison between SUPREME and MOGAT was performed, as we found that MOGONET is incompatible with eight datatypes. The macro-F1 scores of 255 ($2^{8}$ − 1) different combinations of eight datatypes are calculated (Table 1) and plotted using violin plots in Figure 2b. MOGAT outperforms with an average score of 0.797 compared to 0.686 for SUPREME.

For the METABRIC cohort, macro-F1 scores of 63 ($2^{6}$ − 1) combinations of six datatypes were calculated (Figure 2c). We observed that MOGAT has higher macro-F1 scores with an average of 0.745 compared to 0.566 and 0.732 for MOGONET and SUPREME, respectively. Overall, our proposed approach, MOGAT, outperforms MOGONET by 32% to 46% and SUPREME by 2% to 16% in cancer subtype prediction in different combinations of multi-omics data, supporting our hypothesis.

#### Omics-Specific Contribution in Prediction

We also investigated the contribution of each omics data type in cancer subtype prediction, and the results are shown in Table 2. Eight combinations were considered for eight different datatypes, where each combination constitutes all datatypes except the one whose contribution will be investigated. The last row shows the performance of MOGAT using all data types. The performance was estimated in terms of accuracy, weighted-F1 score, and macro-F1 score. The performance metrics were calculated from ten runs, and their mean and standard deviations are reported in Table 2. We observed that MOGAT performs better using all types of data than the other eight combinations where one data type is absent, which means that each data type contributes toward subtype prediction. It is noticeable that the performance without data type EXP (i.e., mRNA expression) is the lowest compared to the performance using all data types, with accuracy 0.837 vs. 0.861, weighted-F1 score 0.831 vs. 0.861, and macro-F1 score 0.766 vs. 0.826. This means that mRNA expression contributes the most toward subtype prediction. On the other hand, the data type MIR (i.e., miRNA expression) has the lowest contribution towards subtype prediction.

For METABRIC, it was also observed that all datatypes produce the highest performance compared to other combinations of datatypes where only one type of data is excluded. Unlike TCGA, the combination without datatype CLI (clinical) has the lowest compared to performance using all datatypes, with accuracy 0.754 vs. 0.791, weighted-F1 0.755 vs. 0.790, and macro-F1 0.736 vs. 0.762, meaning it has the highest contribution towards prediction. On the other hand, COE has the lowest contribution towards subtype prediction.

### 2.2. Visualization

To investigate whether the embeddings can capture the underlying insights of the data, we used principal component analysis (PCA) [18] and tSNE [19], which are dimensionality reduction techniques commonly used to reduce the high-dimensional data to a lower dimensional space to visualize the data. Figure 3 shows the PCA and tSNE plots for the learned GAT embeddings, with their counterpart raw feature matrix for TCGA-BRCA and METABIRC. We observed that for both cohorts, the embeddings learned the underlying structure of the data. The PCA and tSNE plots of GAT embeddings make it easier to tell the difference between groups of points that represent different types of breast cancer than the raw feature matrix plots that were used to train the GATs. 

### 2.3. Survival Analysis to Evaluate GAT Embeddings

Survival analysis was performed for TCGA-BRCA and METABRIC using raw features and GAT embeddings separately, following the methods described in Section 4.14, to evaluate the performance of our framework, MOGAT. The high-risk group contains patients with a risk score higher than the median, and the low-risk group has a score less than or equal to the median. The Kaplan-Meier curves using raw features and GAT embeddings are shown in Figure 4. It is observed that, in both cases, the difference in survival between high-risk and low-risk groups is significant. However, GAT embeddings can distinguish the high-risk and low-risk groups with higher significance than the raw features, as denoted by the log-rank *p*-value (2.10 × 10^−30^ vs. 7.85 × 10^−3^ for TCGA-BRCA and 2.03 × 10^−27^ vs. 2.46 × 10^−16^ for METABRIC).

## 3. Discussion

The hypothesis of the present study was that the attention-based graph neural network would provide better embeddings compared to the graph convolutional neural network (GCN). Our proposed MOGAT model for integrating multi-omics data based on graph attention network (GAT) provides better embeddings and performs better than the GCN-based approaches, such as MOGONET and SUPREME. MOGAT outperforms MOGONET by 32% to 46% and SUPREME by 2% to 16% in cancer subtype prediction in different combinations of multi-omics data, thus supporting our hypothesis.

The rationale for proposing GAT in multi-omics integration is that it has the built-in advantage of employing attention mechanisms to weigh neighbors’ contributions, allowing each node to adaptively focus on its most informative neighbors during message passing. This can lead to better model generalization. In our study, nodes are patients. For example, if a node represents a patient with the basal subtype of breast cancer and has five neighbors, of which two are the basal subtype, it would be realistic to assign more attention to neighbors with the basal subtype than other subtypes. On the other hand, attention mechanisms introduce additional computational overhead. For large-scale graphs, this can make training and inference slower compared to simpler aggregation methods.

The GAT is a specific type of graph neural network (GNN) that utilizes attention mechanisms to dynamically weigh the importance of neighboring nodes during message passing. This means that GNN represents a broad family of neural network architectures designed for graph-structured data. This family includes various architectures and mechanisms, such as graph convolutional networks (GCNs), spectral-based GNNs, message-passing neural networks (MPNNs), and, of course, GATs, among others. Different GNN models have different mechanisms for aggregating information from neighbors. For instance, GCNs use a fixed weight averaging scheme, while MPNNs can employ more general message and update functions.

In the present study, we included eight types of data, including mutations, copy number alterations, mRNA expression, lncRNA expression, miRNA expression, co-expression eigengenes, DNA methylation, and clinical data. Note that we did not include protein expression for analysis. The reason is that it might reflect the same information as mRNA expression since mRNAs are translated into amino acids to form proteins. As a result, both mRNA expression and protein expression might generate similar patient similarity networks. There is already a concern that analysis by integrating seven omics might be an overkill. Adding protein expression could bemore overkill. Whether analysis using too many omics is an overkill deserves further investigation, which will be addressed in our future work.

The study presented in this work has some limitations that need to be addressed in future work. It was restricted to using only the TCGA-BRCA and METABRIC cohorts to predict its five different cancer subtypes. To check the efficacy of the proposed methodology, we will consider subtypes of other cancers in a pan-cancer analysis as well as other diseases, such as Alzheimer’s and Parkinson’s.

The current study also presents opportunities for future research. The framework used in this study utilized a patient similarity network as the input graph, with each node representing a patient. However, it is possible to reorganize the framework so that each node represents a gene, making the task of the graph neural network a graph classification instead of a node classification. While some existing methods follow this approach [5,8], it limits the number of omics data that can be incorporated as node features. For instance, when using genes as nodes, gene expression, somatic mutation, copy number variation, and DNA methylation can be incorporated, but not lncRNAs, miRNAs, and co-expression eigengenes due to the absence of a one-to-one association with genes. To address this, separate graph attention networks with different network and node features are required to incorporate these additional omics data.

Our framework is based on supervised machine learning, where the number of subtypes must be known beforehand. An unsupervised machine learning-based framework would allow for scenarios where the number of subtypes is not known. We envisage the integration of MOGAT with clustering or other unsupervised learning methods to tackle such scenarios.

## 4. Materials and Methods

### 4.1. Dataset Preparation and Preprocessing: TCGA

To develop and investigate the MOGAT approach, we downloaded omics and clinical data for breast invasive carcinoma (BRCA) from the GDC portal (https://portal.gdc.cancer.gov, accessed on 16 December 2022) of The Cancer Genome Atlas (TCGA). The RNAseq gene (mRNA, miRNA, and lncRNA) expression, DNA methylation, copy number variation, simple nucleotide variation, and clinical data were collected for this cohort. Table 3 summarizes the processed omics data with the number of features in different preprocessing steps. The preparation and preprocessing of different types of data are outlined in Figure 5.

#### 4.1.1. Features Based on Clinical Data (CLI Features)

The clinical data consists of age, race, neoplasmic cancer status, tumor stage, menopause status, estrogen receptor status, progesterone receptor status, and HER2 receptor status. The variables were one-hot encoded except for age. Finally, the number of features remained at 31, as shown in Appendix A.

#### 4.1.2. Features Based on Copy Number Alterations (CNA Features)

Appendix A provides the details of processing CNA features from copy number segment mean to a gene-centric matrix with an example. Copy number alteration (CNA) data came as TSV files (Appendix A), each for one sample. Each row of the TSV file corresponds to a genomic coordinate for which the copy number alterations were observed, and the corresponding segment mean value is defined as: ${log}_{2}\left(CopyNumber/2\right)$. These files were combined into a single TSV file for all patients containing 1,268,167 rows. Then, CNTools [20] was used to obtain gene-centric values from the segmented copy number variation data, as explained in Appendix A, and the number of genes was 28,918.

#### 4.1.3. Features Based on Gene Co-Expression (COE Features)

For generating a gene co-expression network, WGCNA [21] was used. The optimal soft-threshold power *β* was selected by evaluating its effect on the scale-free topology model fit $R^{2}$. From the beta values from 1 to 30, 4 was the lowest while maintaining the high $R^{2}$ values (threshold 0.90) (Appendix A). The adjacency matrix was first transformed into a topological overlap matrix (TOM) -based similarity matrix for module detection. Then, it was converted into a TOM-based dissimilarity matrix by subtracting from unity (1). This matrix was used as a distance metric to perform the average linkage hierarchical clustering algorithm, which outputs a dendrogram. Then, the dynamic tree cut [22] method was performed for branch cutting to generate network modules. This process identified 40 modules. Appendix A, shows the number of genes for each of the 40 modules identified by WGCNA. The values of eigengenes for each patient are provided in Appendix A.

#### 4.1.4. Features Based on mRNA Expression (EXP Features)

RNAseq expression data contain the expression of 60,660 genes (including mRNAs, miRNAs, and lncRNAs), from which expression values of 19,962 mRNAs were isolated. The expression values were in FPKM (fragments per kilobase of transcript per million mapped reads). We employed three different approaches in sequence to reduce the original high-dimensional feature space to a meaningful low-dimensional space. First, some of the original features have very small values, such as FPKM ≤ 1 for many samples, which do not carry signals for analysis. The mRNAs are filtered out if their expression values do not meet the threshold of FPKM ≥ 1 in ≥15% of samples (as used in [11]), which resulted in 13,503 mRNAs. Second, these mRNAs were used to perform differential gene expression analysis using DESeq2 [23]. After using the criteria of an adjusted *p*-value ≤ 0.01, the number of remaining mRNAs was 5343, which we referred to as cleaned features, Table 3. Third, a well-known random forest-based feature selection package, BORUTA [24], was used to identify 1000 significant mRNAs. The first 3 rows of Table 3 summarize the feature selection results.

#### 4.1.5. Features Based on lncRNA Expression (LNC Features)

Applying the similar preprocessing and feature selection approaches used for mRNA, we selected 500 significant lncRNAs and corresponding expression values from the original dataset of 60,660 gene expressions.

#### 4.1.6. Features Based on miRNA Expression (MIR Features)

The miRNA expression data were in reads per million (RPM) units. The miRNAs were filtered out if their expression values did not meet the threshold of RPM ≥1 in ≥ 30% of samples, which resulted in 393 miRNAs. Then, differential gene expression analysis using an adjusted *p*-value ≤ 0.01 resulted in 306 miRNAs. The BORUTA feature selection was not used for MIR since the number of features is already low.

#### 4.1.7. Features Based on DNA Methylation (MET Features)

HumanMethylation 27 k (HM27) and HumanMethylation 450 k (HM450) data were collected for DNA methylation. The samples and probes were 343 and 27,578 for HM27 and 895 and 485,577 for HM450, respectively. After combining both datasets by keeping the same probes, the samples and probes were 1238 and 25,978, respectively.

#### 4.1.8. Features Based on DNA Mutation (MUT Features)

For simple nucleotide mutation data, there were 992 samples, and each sample contained a different set of genes for which one or more mutations were observed. The sample mutation data were converted into a vector of genes, where 1 signifies a mutation occurred and 0 signifies no mutation. The size of this vector is the union of all the genes from all samples, which is 16,662.

### 4.2. Original Features to Cleaned Features

Original features with gene expression values (i.e., EXP, LNC, and MIR features) have very small values, such as FPKM ≤ 1 or RPM ≤ 1 for many samples, which do not carry signals for analysis. Those features were filtered out. Then, differential gene expression analysis was conducted to determine the significant features, which we referred to as “Cleaned Features”. The numbers in blue in the “Cleaned Features” row in Table 3 represent the cleaned EXP, LNC, and MIR features. The number of features from other data types remained the same as “Original Features” since they do not require any filtering.

### 4.3. Cleaned Features to Selected Features

Among the cleaned features in Table 3, copy number alteration (CNA), mRNA expression (EXP), lncRNA expression (LNC), DNA methylation (MET), and mutation (MUT) are high-dimensional compared to sample size. Focusing on a subset of significant features can reduce potential noise and overfitting often associated with high-dimensional data. Thus, the Boruta package [24], a feature selection method based on the random forest algorithm, was used for the feature selection process. Without this feature selection, node features would have very high dimensions (~29 K for CNA, ~5 K for EXP, ~3 K for LNC, ~26 K for MET, and ~17 K for MUT; in total, ~80 K features for TCGA-BRCA) as opposed to the sample size (920 patients). This step was not used for co-expression (COE), miRNA expression (MIR), and one-hot encoded clinical features (CLI), as they did not have many features like their other counterparts. The numbers in red in the “Selected Features” row of Table 3 are features selected by Boruta. The other three types of features remained the same as “Cleaned Features”.

### 4.4. Dataset Preparation and Preprocessing: METABRIC

The METABRIC (Molecular Taxonomy of Breast Cancer International Consortium) [25] cohort data were collected from cbiportal.com. The summary of the preparation and preprocessing of METABRIC data is given in Table 4. The repository contained clinical (CLI), copy number alterations (CNA), mRNA expression (EXP), DNA methylation (MET), and somatic mutation (MUT) data. For the clinical dataset, age at diagnosis, menopause status, estrogen receptor status, progesterone status, HER2 status, and ethnicity were considered for analysis. The variables were converted into a one-hot vector, except for age. Finally, the number of features remained at 14. The list of clinical features is given in Appendix A. The other datasets were already preprocessed into a gene matrix, unlike the TCGA-BRCA data from the GDC portal.

The co-expression eigengenes (COE) as the features from the gene co-expression network were generated using WGCNA, following a similar procedure for the TCGA-BRCA cohort. Appendix A shows that 6 was the lowest value of soft-threshold power *β* while maintaining the high scale-free topology model fit $R^{2}$ value at 0.90. There were 49 co-expression modules, and 49 features (module eigengene, which is the 1st PCA component using expression of genes in a module) were found. Appendix A shows the number of genes in each of the 49 modules identified by WGCNA. The values of eigengenes for each patient are provided in Appendix A.

### 4.5. Duplicate Data Handling

It was observed that some patients have more than one sample in different omics datasets for the TCGA-BRCA cohort. For example, a patient with ID: TCGA-A7-A0DB contains three samples of mRNA expression, miRNA expression, and mutation data. Therefore, to make sure each patient has one corresponding sample, an extra preprocessing step was taken. For mRNA expression, miRNA expression, co-expression, DNA methylation, and copy number alterations data, the multiple samples corresponding to single patients were replaced with one sample by taking the average value for each feature. Since the mutation data are in binary (0/1), the Boolean OR operation was performed instead of the average for the patients with more than one sample.

### 4.6. Missing Data Handling

In multi-omics integration analysis, data could be missing both across and within omics [26]. In multi-omics study designs, it is common for individuals to be represented for some omics layers but not all, which results in across-omics missing data. The same is true for both datasets used for breast cancer, one from TCGA and the other from METABRIC. For TCGA breast cancer data, the number of samples for eight omics varies from 969 for mutation to 1097 for DNA methylation (row “Unique Tumor Samples” of Table 3), and the six omics for METABRIC vary between 1418 for gene expression and 2509 for DNA methylation (row “All Samples” in Table 4). This means that both datasets have across-omics missing data. We used the common samples across omics to avoid bias due to the across-omics missing data, which are 920 and 1372 for the TCGA and METABRIC datasets, respectively.

The major issue with within-omics missing data is that true zeros (representing the true gene expression levels, for example) are mingled with dropout zeros (representing the actual missing data) [27], which is altogether a different topic and beyond the scope of this study. The multi-omics integration pipelines—MOGONET and SUPREME—we are comparing did not consider the handling of within-omics missing data.

### 4.7. Normalizing Feature Values

The selected feature values in categories CNA, EXP, LNC, MIR, and MET are z-score normalized. The CLI (one-hot encoded), COE features (eigengenes), and MUT features (binary) do not require normalization.

### 4.8. Final Set of Patients for Analysis

Our objective is to integrate the most available number of omics data and to investigate the effects of each data type on a patient’s outcome. However, different omics datasets contain different numbers of samples. The union of samples would lead to some patients not having features from all types of omics data; thereby, incorporating them would not meet the study objective. Thus, we used the intersection of samples from all types of data, which resulted in 920 and 1372 tumor samples common in all types for the TCGA-BRCA and METABRIC cohorts, respectively. Therefore, these common tumor samples were used to create the patient similarity networks (last row in Table 3 and Table 4) and the feature matrices.

### 4.9. Network and Feature Matrix Construction

The input to the graph attention network is a network and the feature matrix for the nodes in the given network. In this case, the training would be in single mode, where each sample corresponds to a node in the network, and the whole network represents a data type (gene expression, mutation, etc.).

#### 4.9.1. Network Construction

In the present study, a network is a patient similarity network, where a node represents a patient and an edge represents the similarity between two patients. The similarity matrix was created for each data type before constructing the corresponding patient similarity network. The similarity was computed using Pearson’s correlation for mRNA expression, miRNA expression, lncRNA expression, co-expression features, copy number alterations, and DNA methylation. The Jaccard similarity was used instead of Pearson’s correlation network for mutation data, as it is binary. The Gower metric [28] was used to compute patient similarity using clinical data since it combines categorical and continuous features. Then, from each similarity matrix, the top 3 similar samples for each sample were selected as edges to construct the similarity network.

#### 4.9.2. Feature Matrix Construction

Each row of the feature matrix represents the feature vector for a node (here, a patient), which is the concatenation of all types of features for that patient coming from eight types (for the TCGA cohort) or six types (for the METABRIC cohort) of data. Note that only the selected features from each data type are concatenated, which resulted in a feature matrix of 920 × 3577 and 1372 × 3345 for the TCGA and METABRIC cohorts, respectively.

### 4.10. Interaction between Omics Data

The interactions between different omics types are considered during node feature engineering. The patient similarity network was constructed based on omics-specific data, but the node features contain features from all the omics data for each node or patient. While individual graph attention networks operate independently, they do so on an integrated foundation based on node features from all omics data. The post-analysis concatenation represents an integration of these learned representations, not a simplistic merger of isolated data types.

### 4.11. Graph Attention Network

The utilized GAT model is based on the idea of the self-attention mechanism, where embeddings are created from eight different types of data (Table 3) with the assumption that samples with similar characteristics (such as gene expression or DNA methylation) are likely to have similar disease outcomes and are, therefore, related to each other. However, not all related samples should be given equal importance. Some samples may have a greater impact on the prediction or clustering of a target sample, which cannot be accurately determined by similarity metrics. To address this, the GAT model assigns varying levels of attention to a target sample’s neighboring samples, allowing it to capture the significance of each one.

For each data type, let *n* be the number of samples or patients and *m* be the number of features (concatenated from different omics types). The input feature matrix is given by $X=\left[x_{1},x_{2},\;\ldots ,\;x_{n}\right]$, where $x\in \;R^{1\times m}$ represents a sample feature vector. While generating the embedding of sample $x_{i},$ the attention given to it from its neighbor $x_{j}$ can be calculated as:(1)$$c_{ij}=LeakyReLU(a^{T}[Wx_{i}\;||\;Wx_{j}])$$
where *W* $\in \;R^{p\times m}$ and $a\in \;R^{2p\times 1}$ are learnable weight parameters, shared across all samples and p is the embedding size; || symbol denotes the concatenation of two vectors; and *LeakyReLU* is the non-linear activation function. $c_{ij}$ describes the importance of sample j’s feature to sample i. We then normalize attention coefficients by applying a SoftMax function:(2)$$\alpha_{ij}=Softmax\left(c_{ij}\right)=\frac{\exp\left(c_{ij}\right)}{\sum_{k\in N_{i}}\exp\left(c_{ik}\right)}$$
where $N_{i}$ is the set of neighboring nodes of sample i. With the normalized attention coefficients being the weights, a linear combination of input features is used as the output representation for each data sample. Formally, we have:(3)$$h_{i}=\sum_{j\in N_{i}}\alpha_{ij}Wx_{i}$$
where $h_{i}$ is the output representation of sample i.

### 4.12. Training GAT

For training the GATs, the architecture remained the same for all the datatypes, consisting of two graph attention layers. The hidden layer dimension for each GAT model and learning rate were selected based on grid search-based hyperparameter tuning. The ranges of values for hyperparameters are listed in Table 5. For LeakyReLU, the hyperparameter called negative input slope, α, was used as 0.2 following [16].

### 4.13. Classification

Embeddings generated after training the GATs were concatenated and used as input for classification. A multi-layer perceptron (MLP) was used to classify breast cancer subtypes. The architecture, learning rate, and number of epochs for MLP were selected based on a randomized grid search. The range of values is listed in Table 5.

The classification metrics, including accuracy, weighted-F1 score, and macro-F1 score, were estimated to evaluate the performance of the MOGAT model.

### 4.14. Implementation

All experiments were conducted on a Linux machine with 8 NVIDIA A100 GPUs, each with 40 GB of memory. The software environment was CUDA 11.6 and Driver Version 520.61.05. We used Python 3.9.13 and Pytorch 1.12.1 to construct our project. Other packages and their versions are available in the GitHub repository.

### 4.15. Survival Analysis

Survival analysis was performed using raw features (concatenated selected features after cleaning and feature selection) and GAT embeddings separately. Table 6 shows the number of raw features and GAT embeddings at different stages of survival analyses. The initial numbers of raw features were 3577 and 4335 for TCGA-BRCA and METABRIC, respectively. The initial numbers of embeddings were 4096 (8 × 512) and 3072 (6 × 512). For the TCGA-BRCA cohort, LASSO regression with overall survival as output reduced the number of raw features and embeddings to 276 and 2247, respectively. The regularizing factor α for LASSO was selected using a Grid SearchCV approach. The list of values for α used in Grid Search and the optimized value are given in Table 5.

Next, a multivariate Cox proportional hazard (Cox-PH) regression analysis [29] was conducted using the selected features in the previous step. This technique examines the influence of multiple variables on the time it takes for a specific event to occur, in this case, death. In the Cox regression model, the coefficients of predictor variables (raw features or embeddings) are related to hazard, i.e., risk of death. A positive coefficient indicates a worse prognosis, and a negative coefficient indicates a protective effect of the variable with which it is associated. The exponent of its coefficient gives the hazard ratio associated with a predictor variable, and the *p*-value shows the significance of the association between the predictor variable (raw feature or embedding) and the risk of death. The significant predictor variables, 57 raw features and 542 embeddings, with a *p*-value < 0.05, were selected, and their Cox coefficients were used to calculate the risk scores using raw features and embeddings, respectively.
(4)$$RiskScore=\sum X_{i}*Coef_{i}$$
where $X_{i}$ is the value of *i*-th predictor (raw feature or embedding) and $Coef_{i}$ is the corresponding coefficient for the predictor obtained from the Cox regression. This Risk Score is used to divide the cohort into low-risk and high-risk groups using the median as the divider. Then, Kaplan–Meier [30] and logrank tests [31] were performed, and hazard ratios were calculated to see if the two groups were significantly distinguishable.

## Figures and Tables

**Figure 1 ijms-25-02788-f001:**
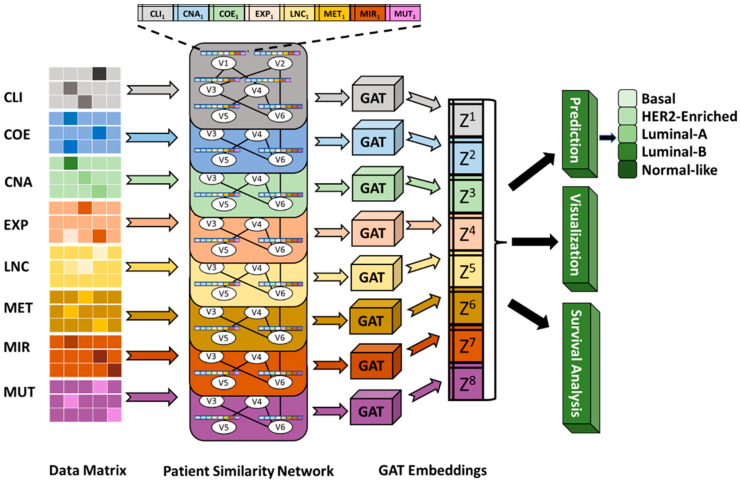
Illustration of the MOGAT framework. The MOGAT framework processes patient similarity networks constructed from eight datatypes, including CLI (clinical), CNA (copy number alteration), COE (co-expression), EXP (mRNA expression), LNC (lncRNA expression), MET (DNA methylation), MIR (miRNA expression), and MUT (simple nucleotide variation). Nodes in each patient similarity network are annotated with features from eight data types. By applying GAT to each patient similarity network, the framework generates embeddings. These embeddings are then used for subtype prediction, visualization, and survival analysis.

**Figure 2 ijms-25-02788-f002:**
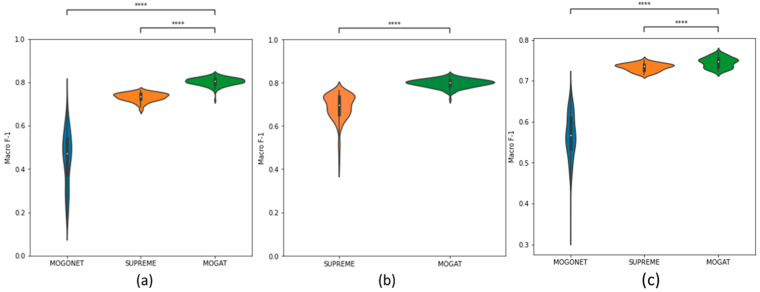
Macro-F1 score to compare MOGAT with state-of-the-art multi-omics integration frameworks, MOGONET and SUPREME. (**a**,**b**) with TCGA-BRCA data and (**c**) with METABRIC data. Comparison between (**a**) MOGONET, SUPREME, and MOGAT using seven ($2^{3}-1$) combinations of three omics data (EXP, MET, MIR); (**b**) SUPREME and MOGAT using 255 ($2^{8}-1$) combinations of eight omics data; (**c**) MOGONET, SUPREME, and MOGAT using 63 ($2^{6}-1$) combinations of six omics data. A pairwise statistical comparison was performed using the Mann-Whitney Wilcoxon test with a two-sided Bonferroni correction. The *p*-value annotations are—****: *p* ≤ 0.0001.

**Figure 3 ijms-25-02788-f003:**
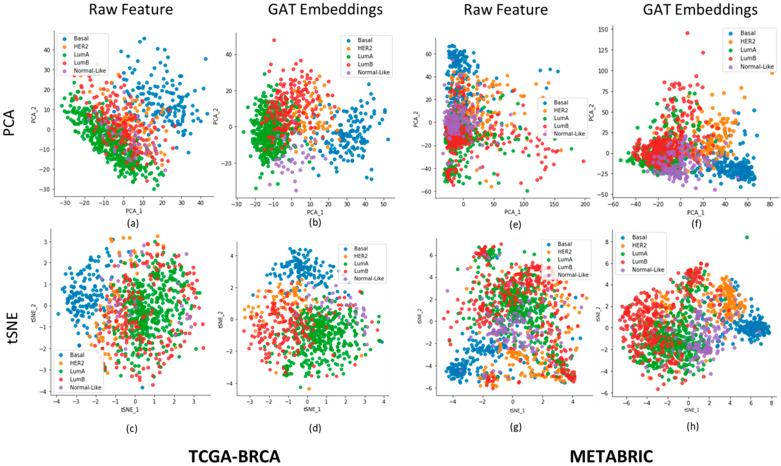
PCA and tSNE plots of breast cancer patients using raw features and embeddings. (**a**–**d**): for TCGA-BRCA. (**e**–**h**): for METABRIC. The X-axis and Y-axis correspond to the first and second components of PCA or tSNE.

**Figure 4 ijms-25-02788-f004:**
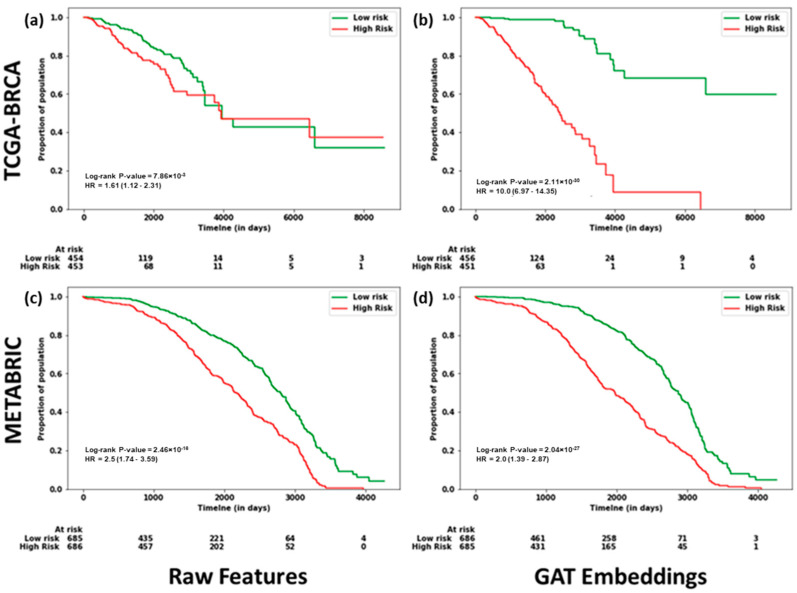
Survival analysis using raw features and GAT embeddings. (**a**,**b**): TCGA-BRCA. (**c**,**d**): METABRIC. Kaplan–Meier curves showing the proportion of population of low-risk and high-risk groups at different observation times. The low-risk and high-risk groups were determined by their risk scores calculated using raw features (**a**,**c**) and GAT embeddings (**b**,**d**).

**Figure 5 ijms-25-02788-f005:**
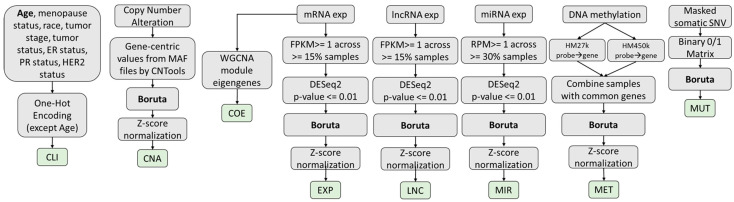
Flowcharts for data preprocessing for each datatype used in the MOGAT framework.

**Table 1 ijms-25-02788-t001:** Macro-F1 score to compare MOGAT with state-of-the-art multi-omics integration frameworks, MOGONET and SUPREME. The first two rows correspond to TCGA-BRCA with three omics (as shown in Figure 2a) and eight omics (as shown in Figure 2b) data. The last row corresponds to METABRIC with six omics data (as shown in Figure 2c). For each scenario, the average with standard deviation is listed. The last column shows the percentage improvement of the proposed MOGAT over MOGONET and SUPREME.

Data	Model	Avg ± SD	Improvement
**TCGA: 3 omics**	**MOGAT**	**0.804 ± 0. 017**	**---**
	SUPREME	0.732 ± 0.019	10%
	MOGONET	0.550 ± 0.145	46%
**TCGA: 8 omics**	**MOGAT**	**0.797 ± 0.019**	**---**
	SUPREME	0.686 ± 0.062	16%
**METABRIC: 6 omics**	**MOGAT**	**0.745 ± 0.012**	**---**
	SUPREME	0.733 ± 0.008	2%
	MOGONET	0.566 ± 0.056	32%

**Table 2 ijms-25-02788-t002:** Contribution of individual omics datatype. Performance of MOGAT with different combinations of datatypes in terms of accuracy, weighted-F1, and macro-F1 for TCGA-BRCA and METABRIC. The last row contains the result of integrating/embedding all data types. Each of the other rows contains embeddings from every datatype except the one noted in the first column.

	TCGA-BRCA			METABRIC		
Used Embeddings	Accuracy	Weighted F1	Macro F1	Accuracy	Weighted F1	Macro F1
All except CLI	0.842 ± 0.023	0.840 ± 0.026	0.790 ± 0.038	0.754 ± 0.02	0.755 ± 0.02	0.736 ± 0.02
All except CNA	0.837 ± 0.03	0.832 ± 0.041	0.775 ± 0.088	0.784 ± 0.012	0.782 ± 0.013	0.753 ± 0.016
All except COE	0.856 ± 0.02	0.853 ± 0.024	0.792 ± 0.044	0.790 ± 0.008	0.788 ± 0.008	0.759 ± 0.009
All except EXP	0.837 ± 0.012	0.831 ± 0.016	0.766 ± 0.041	0.775 ± 0.012	0.772 ± 0.012	0.748 ± 0.014
All except LNC	0.848 ± 0.023	0.849 ± 0.025	0.792 ± 0.043	N/A	N/A	N/A
All except MET	0.850 ± 0.013	0.847 ± 0.016	0.802 ± 0.056	0.782 ± 0.012	0.779 ± 0.012	0.75 ± 0.013
All except MIR	0.859 ± 0.013	0.856 ± 0.01	0.814 ± 0.029	N/A	N/A	N/A
All except MUT	0.848 ± 0.01	0.848 ± 0.018	0.798 ± 0.024	0.778 ± 0.024	0.776 ± 0.025	0.751 ± 0.027
**All Datatypes**	**0.861 ± 0.024**	**0.861 ± 0.03**	**0.826 ± 0.069**	**0.791 ± 0.009**	**0.790 ± 0.01**	**0.762 ± 0.013**

Red: lowest performance among All except single omics; Purple: highest performance among All except single omics; Bold: all omics provide best performance.

**Table 3 ijms-25-02788-t003:** The summary of each datatype for the TCGA-BRCA cohort. Row 1 (Original Features): number of original features, Row 2 (Cleaned Features): number of features after cleaning by filtering, Row 3 (Selected Features): number of features after applying the Boruta feature selection approach, Row 4 (All Tumor Samples): Number of tumor samples, including duplicates, Row 5 (Unique Tumor Samples): number of tumor samples after removing duplicates, Row 6 (Common Samples): subtype distribution of the tumor samples common across all datatypes, and Row 7 (Network): number of nodes and edges for the patient similarity network for each datatype.

Datatype	CLI	CNA	COE	EXP	LNC	MET	MIR	MUT
Original Features	31	28,918	40	19,962	16,901	25,978	1881	16,662
Cleaned Features	31	28,918	40	5343	3398	25,978	306	16,662
Selected Features	31	500	40	1000	500	1000	306	200
All Tumor Samples	1089	1106	1212	1212	1212	1107	1069	992
Unique Tumor Samples	1089	1096	1076	1076	1076	1097	1057	969
Common Samples	920 Samples; Basal: 158 (17.24%) HER2: 73 (8.02%) LumA: 467 (50.65%) LumB: 188 (20.39%) NL: 34 (3.68%)							
Network (Nodes, Edges)	(920, 2398)	(920, 2346)	(920, 2122)	(920, 2391)	(920, 2218)	(920, 2675)	(920, 2108)	(920, 2752)

Note: CLI: clinical, CNA: copy number alteration, COE: co-expression, EXP: gene expression, LNC: lncRNA expression, MET: DNA methylation, MIR: miRNA expression, MUT: simple nucleotide mutation. LumA: Luminal A, LumB: Luminal B, NL: Normal-like. Blue: Number of features after cleaning; Red: Number of features after feature selection using Boruta.

**Table 4 ijms-25-02788-t004:** The summary of each datatype for the METABRIC cohort. Row 1 (Original Features): number of original features, Row 2 (Selected Features): number of features after applying the Boruta feature selection approach, Row 3 (All Samples): number of samples (no duplicates), Row 4 (Common Samples): subtype distribution of the tumor samples common across all datatypes, and Row 5 (Network): number of nodes and edges for the patient similarity network for each datatype.

Datatype	CLI	CNA	COE	EXP	MET	MUT
Original Features	14	22,544	49	24,368	13,188	173
Selected Features	14	1003	49	1048	1058	173
All Samples	2508	1905	1905	1418	2509	2174
Common Samples	1372 Samples; Basal: 218 (15.89%); HER2: 181 (13.19%); LumA: 500 (36.44%); LumB: 335 (24.42%); Normal-like: 138 (15.89%)					
Network (Nodes, Edges)	(1372, 5055)	(1372, 4853)	(1372, 4593)	(1372, 5023)	(1372, 5329)	(1372, 4765)

**Table 5 ijms-25-02788-t005:** Hyperparameter Tuning for GAT, MLP, and LASSO. The type of hyperparameter and their ranges of values used for tuning are provided. Optimized hyperparameter values for GAT and LASSO are bolded for TCGA-BRCA. For MLP tuning, it has 255 sets of optimized hyperparameter values, one for each combination of multi-omics data.

Hyperparameters	Values
GAT hidden layer dimensions	[128, 256, **512**, 1024]
GAT Learning Rate	[0.01, **0.001**, 0.0001]
GAT # of epochs GAT # of heads	[100, **200**, 500] [**1**, 2, 4, 8]
MLP learning rate	[0.1, 0.01, **0.001**, 0.0001, 0.00001]
MLP hidden layer architecture	[(32), (64), (128), (256), (512), (32, 32), **(64, 32)**, (128, 32), (256, 32)]
MLP # of epochs	[200, 500, **1000**, 1500]
LASSO regularizing factor α	[0.001, 0.002, 0.005, 0.01, 0.05, **1.0**]

**Table 6 ijms-25-02788-t006:** The number of variables (Raw features and GAT embeddings) that remained at each step of the survival analysis process. Initial number of raw features is the sum of the reduced set of features from eight or six different data types (Row: “Selected Features” from Table 3 and Table 4).

	TCGA-BRCA		METABRIC	
Item	Raw	Embeddings	Raw	Embeddings
Initial	3577	4096	4335	3072
After LASSO	276	2247	21	21
After Cox-PH	57	542	8	6

## Data Availability

The code and data required for training the model are provided in the GitHub link: https://github.com/MezbahJUCSE39/MOGAT.

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
