# Peer review of "MOGAT: A Multi-Omics Integration Framework Using Graph Attention Networks for Cancer Subtype Prediction"

_ijms, 2024, doi:10.3390/ijms25052788_

Round 1

Reviewer 1 Report

Comments and Suggestions for Authors

In the current study, authors construct a platform MOGAT for multiomic integration. It is an important work and the authors did a great job presenting a new platform.

However, there are some minor suggestions. 

  1. Besides TCGA-BRCA and METABRIC, can author extract data from one more database and do the comparison with previous multiomic integration platform, MOGONET and SUPREME
  2. Does author consider integrating protein expression data into MOGAT? Protein expression is important and maybe more important and accurate that mRNA expression data.
    If not yet, could author discuss in the discussion? 
  3. Can author discuss more advantage and pitfall of Graph Attention Network (GAT)? This discussion is lack in the discussion area. 
  4. In discussion, also more discussion and comparison between GAT and GNN (or other methods)?

Comments on the Quality of English Language

The English writing is okay. 

Author Response

Response to Reviewer 1 Comments

After reading the comments provided by the reviewers on our article submitted to MDPI IJMS journal, we were excited to learn of the reviewers’ enthusiasm for our proposed study, “MOGAT: A Multi-Omics Integration Framework Using Graph Attention Networks for Cancer Subtype Prediction.” We were delighted that the reviewers agreed that (a) “It is an important work, and the authors did a great job presenting a new platform,” and (b) “The model is described in great detail, as well as the initial data and their preparation.”

The scope or objective of this study is to show that the attention-based graph neural network performs better than the convolutional graph neural network in multi-omics integration-based analysis. To prove our hypothesis, we used two widely used well-known datasets for breast cancer, which are considered benchmark datasets for breast cancer: One from TCGA and the other from METABRIC. We provide our response to reviewer comments those are within the scope of our study. Addressing the other comments (not within the scope of this article) leads to ideas for new articles with new experiments. We are greatly appreciative of the out-of-the-scope comments or suggestions leading to new ideas for new articles. The out-of-the-scope comments are summarized as future works in the revised version of the article.

Reviewer 1

In the current study, authors construct a platform MOGAT for multiomic integration. It is an important work, and the authors did a great job presenting a new platform. However, there are some minor suggestions.

We thank the reviewer for providing a very positive and enthusiastic comment on our proposed computational framework, MOGAT for multi-omics integration, by saying that “It is an important work, and the authors did a great job presenting a new platform.”

  1. Besides TCGA-BRCA and METABRIC, can author extract data from one more database and do the comparison with previous multiomic integration platform, MOGONET and SUPREME.

Thank you for the suggestion to use another set of data. To address this issue, we need to add another cancer data, for example, lung cancer or liver cancer. We need to do all the preprocessing to generate the network then do embedding using GAT and finally do the classification using embedding. But it is going to take considerable time and addressing this comment can lead to a pan-cancer analysis using multi-omics integration, which is our future plan.

Manuscript Update (“Discussion and Future Directions” Section):

To check the efficacy of the proposed methodology, we will consider subtypes of other cancer in a pan-cancer analysis as well as other diseases, such as Alzheimer's and Parkinson’s.

  1. Does author consider integrating protein expression data into MOGAT? Protein expression is important and maybe more important and accurate that mRNA expression data. If not yet, could author discuss in the discussion?

Yes, the reviewer is correct that we did not use protein expression for analysis. There is already a concern that analysis by integrating eight omics might be an overkill. Adding protein expression may not improve the model performance at all since mRNA and protein expression reflect the similar information and they might generate the same/similar patient similarity networks. Whether analysis using too many omics is an overkill deserves further investigation, which will be addressed in our future work.

Manuscript Update (“Discussion and Future Directions” Section):

In the present study, we included eight types of data, including mutation, copy number alterations, mRNA expression, lncRNA expression, miRNA expression, co-expression eigengenes, DNA methylation, and clinical data. Note that we did not include protein ex-pression for analysis. The reason is that it might reflect the same information as mRNA expression since mRNAs are translated to amino acids to form proteins. As a result, both mRNA expression and protein expression might generate similar patient similarity networks. There is already a concern that analysis by integrating seven omics might be an overkill. Adding protein expression could be way more overkill. Whether analysis using too many omics is an overkill deserves further investigation, which will be addressed in our future work.

  1. Can author discuss more advantage and pitfall of Graph Attention Network (GAT)? This discussion is lack in the discussion area.

We are thankful to the reviewer for pointing this shortcoming. We updated the manuscript adding the following paragraph in Discussion Section.

Manuscript Update (“Discussion and Future Directions” Section):

The rationale of proposing GAT in multi-omics integration is that it has built-in advantage of employing attention mechanisms to weigh neighbors' contributions, allowing each node to adaptively focus on its most informative neighbors during message passing. This can lead to better model generalization. In our study, nodes are patients. For example, if a node represents a patient with Basal subtype of breast cancer and has five neighbors of which two are Basal subtype, it would be realistic to assign more attention to neighbors with basal subtype than other subtypes. On the other hand, attention mechanisms introduce additional computational overhead. For large-scale graphs, this can make training and inference slower compared to simpler aggregation methods.

  1. In discussion, also more discussion and comparison between GAT and GNN (or other methods)?

We very much appreciate the reviewer’s comment on inclusion of more discussion and comparison between GAT and GNNs, which will improve the quality of the paper. The following paragraph is added in Discussion Section.

Manuscript Update (“Discussion and Future Directions” Section):

The GAT is a specific type of Graph Neural Network (GNN) that utilizes attention mechanisms to dynamically weigh the importance of neighboring nodes during message passing. This means that GNN represents a broad family of neural network architectures designed for graph-structured data. This family includes various architectures and mechanisms, such as Graph Convolutional Networks (GCNs), Spectral-based GNNs, Message Passing Neural Networks (MPNNs), and, of course, GATs among others. Different GNN models have different mechanisms for aggregating information from neighbors. For instance, GCNs use a fixed weight averaging scheme, while MPNNs can employ more general message and update functions.

Reviewer 2 Report

Comments and Suggestions for Authors

This paper focuses on the construction of a neural network model for predicting breast cancer subtypes based on multiple omics data. The model is described in great detail, as well as the initial data and their preparation. I did not have any questions or comments about the work, except for one -- the work does not provide references to the source codes for both data preparation and model training. I would also like to have a trained model so that it can be applied.

Author Response

Response to Reviewer 2 Comments

After reading the comments provided by the reviewers on our article submitted to MDPI IJMS journal, we were excited to learn of the reviewers’ enthusiasm for our proposed study, “MOGAT: A Multi-Omics Integration Framework Using Graph Attention Networks for Cancer Subtype Prediction.” We were delighted that the reviewers agreed that (a) “It is an important work, and the authors did a great job presenting a new platform,” and (b) “The model is described in great detail, as well as the initial data and their preparation.”

The scope or objective of this study is to show that the attention-based graph neural network performs better than the convolutional graph neural network in multi-omics integration-based analysis. To prove our hypothesis, we used two widely used well-known datasets for breast cancer, which are considered benchmark datasets for breast cancer: One from TCGA and the other from METABRIC. We provide our response to reviewer comments those are within the scope of our study. Addressing the other comments (not within the scope of this article) leads to ideas for new articles with new experiments. We are greatly appreciative of the out-of-the-scope comments or suggestions leading to new ideas for new articles. The out-of-the-scope comments are summarized as future works in the revised version of the article.

Reviewer 2

This paper focuses on the construction of a neural network model for predicting breast cancer subtypes based on multiple omics data. The model is described in great detail, as well as the initial data and their preparation. I did not have any questions or comments about the work, except for one -- the work does not provide references to the source codes for both data preparation and model training. I would also like to have a trained model so that it can be applied.

We are thankful to the reviewer for providing an appreciative comment that “The model is described in great detail, as well as the initial data and their preparation.”

We also thank the reviewer for identifying the shortcoming of not providing “source codes for both data preparation and model training.” To address this issue, we provided a GitHub link in the revised version of the article, https://github.com/mldag2k18/MOGAT, which includes the code and data required for training the model. The data preparation was done using the established packages like BORUTA, CNTools, DESeq2, and WGCNA. The number of input and output features for each preprocessing steps are provided in Table 1 (for TCGA cohort) and Table 2 (for METABRIC cohort). The supplementary files contain the preprocessing information for clinical features (Supplementary 1), CNA features (Supplementary 2), and co-expression features (Supplementary 3) as well as the extracted co-expression features (module eigen values) for TCGA cohort (Supplementary 4) and METABRIC cohort (Supplementary 5). Together, using the information provided in Tables 1 and 2 and Supplementary files (1 through 5), it is possible to reproduce the results using the standard packages BORUTA, CNTools, DESeq2, and WGCNA.

Manuscript Update under “Data Availability Statement”

The code and data required for training the model are provided in the GitHub link, https://github.com/mldag2k18/MOGAT.

Reviewer 3 Report

Comments and Suggestions for Authors

1. The study utilizes data from TCGA for Breast Invasive Carcinoma (BRCA). Concerns may arise regarding the representativeness of this dataset for breast cancer patients in general, as TCGA data may not perfectly mirror the broader population.

2. The study employs various types of omics data, resulting in a high-dimensional feature space. The choice of feature selection methods and the rationale behind it, particularly for reducing dimensionality, should be thoroughly justified.

3. The study involves multiple preprocessing steps, including filtering out features based on thresholds (e.g., FPKM ≥ 1). The impact of these thresholds on the data and the justification for their selection should be discussed.

4. The study does not explicitly mention how missing data is handled. The approach to dealing with missing values is crucial, and the lack of information on this aspect raises concerns about potential bias in the analysis.

5. The GAT model and MLP are used for classification, but details on the hyperparameter tuning process (e.g., the rationale for selecting specific hyperparameter ranges) and the model evaluation methodology (e.g., validation strategy) are not fully provided.

6. The survival analysis involves LASSO regression and Cox Proportional Hazard regression. The reasons for choosing these methods, along with details on the assumptions made and their implications, should be discussed. Additionally, the rationale behind the selection of the threshold (P-value < 0.05) for selecting significant predictor variables is essential.

7. Machine learning is well-known and has been used in previous biomedical studies i.e., PMID: 37112302, PMID: 36642410. Therefore, the authors are suggested to refer to more works in this description to attract a broader readership.

8. The study lacks external validation on an independent dataset. The generalizability of the proposed MOGAT approach should be assessed on datasets other than TCGA-BRCA to ensure its applicability to diverse patient populations.

9. While the study mentions the number of features selected and their significance, the biological and clinical interpretation of these features is crucial.

10. Detailed information on software versions, packages, and settings used in the analysis should be provided to ensure the reproducibility of the study by other researchers.

Comments on the Quality of English Language

English writing should be minor checked.

Author Response

Response to Reviewer 3 Comments

After reading the comments provided by the reviewers on our article submitted to MDPI IJMS journal, we were excited to learn of the reviewers’ enthusiasm for our proposed study, “MOGAT: A Multi-Omics Integration Framework Using Graph Attention Networks for Cancer Subtype Prediction.” We were delighted that the reviewers agreed that (a) “It is an important work, and the authors did a great job presenting a new platform,” and (b) “The model is described in great detail, as well as the initial data and their preparation.”

The scope or objective of this study is to show that the attention-based graph neural network performs better than the convolutional graph neural network in multi-omics integration-based analysis. To prove our hypothesis, we used two widely used well-known datasets for breast cancer, which are considered benchmark datasets for breast cancer: One from TCGA and the other from METABRIC. We provide our response to reviewer comments those are within the scope of our study. Addressing the other comments (not within the scope of this article) leads to ideas for new articles with new experiments. We are greatly appreciative of the out-of-the-scope comments or suggestions leading to new ideas for new articles. The out-of-the-scope comments are summarized as future works in the revised version of the article.

Reviewer 3

  1. The study utilizes data from TCGA for Breast Invasive Carcinoma (BRCA). Concerns may arise regarding the representativeness of this dataset for breast cancer patients in general, as TCGA data may not perfectly mirror the broader population.

Thank you for your thoughtful comment regarding the utilization of TCGA data for our study. Note that we also used another dataset, METABRIC (Molecular Taxonmy of Breast Cancer International Consortium), which represents a broader population.

  1. The study employs various types of omics data, resulting in a high-dimensional feature space. The choice of feature selection methods and the rationale behind it, particularly for reducing dimensionality, should be thoroughly justified.

We appreciate the reviewer’s comment on feature selection methods and the rationale behind it. These points are already enumerated in Section 2.1.4 (Features Based on mRNA Expression (EXP Features)), and Section 2.3 (Cleaned Features to Selected Features). The feature selection results are summarized in Table 1 for BRCA cohort and Table 2 for METABRIC cohort. We updated Section 2.1.4 to bring more clarity.

Manuscript Update:

2.1.4. Features Based on mRNA Expression (EXP Features)

RNAseq expression data contains the expression of 60,660 genes (including mRNAs, miRNAs, and lncRNAs), from which expression values of 19,962 mRNAs were isolated. The expression values were in FPKM (Fragments Per Kilobase of transcript per Million mapped reads). We employed three different approaches in sequence to reduce the original high-dimensional feature space to a meaningful low-dimensional space. First, some of the original features have very small values, such as FPKM ≤ 1 for many samples, which do not carry signals for analysis. The mRNAs are filtered out if their expression values do not meet the threshold of FPKM ≥ 1 in ≥15% of samples (as used in [11]), which resulted in 13,503 mRNAs. Second, these mRNAs were used to perform differential gene expression analysis using DESeq2 [21]. After using the criteria of adjusted P-value ≤ 0.01, the number of remaining mRNAs was 5,343, which we referred to as cleaned features, Table 1. Third, a well-known random forest-based feature selection package BORUTA [22] was used to identify 1,000 significant mRNAs. First 3 rows of Table 1 summarize the feature selection results.

  1. The study involves multiple preprocessing steps, including filtering out features based on thresholds (e.g., FPKM ≥ 1). The impact of these thresholds on the data and the justification for their selection should be discussed.

Yes, the reviewer is right that we used a threshold on FPKM to clean the data. This has been already addressed in the previous comment (Comment 2). Here, we mentioned it again –

First, some of the original features have very small values, such as FPKM ≤ 1 for many samples, which do not carry signals for analysis. The mRNAs are filtered out if their expression values do not meet the threshold of FPKM ≥ 1 in ≥15% of samples (as used in [11]), which resulted in 13,503 mRNAs.

  1. The study does not explicitly mention how missing data is handled. The approach to dealing with missing values is crucial, and the lack of information on this aspect raises concerns about potential bias in the analysis.

We thank the reviewer for this thoughtful comment. In multi-omics integration analysis, data could be missing both within and across omics [1]. But it is not clear, which type of missing data the reviewer is referring.

Across-Omics Missing Data: In multi-omics study designs, it is common for individuals to be represented for some omics layers but not all [1], which results in across-omics missing data. The same is true for both datasets used for breast cancer, one from TCGA and the other from METABRIC. For TCGA breast cancer data, the number of samples for 8 omics varies between 969 for mutation to 1097 for DNA methylation (Row “Unique Tumor Samples” of Table 1), and the 6 omics for METABRIC varies between 1418 for gene expression and 2509 for DNA methylation (Row “All Samples” in Table 2). This means that both datasets have across-omics missing data. Handling across-omics missing data: To avoid the bias due to the across-omics missing data, we used the common samples across omics, which are 920 and 1,372 for TCGA and METABRIC datasets, respectively.

Within-Omics Missing Data: The major issue with within-omics missing data is that true zeros (representing the true gene expression levels, for example) are mingled with dropout zeros (representing the actual missing data) [2], which is altogether a different topic and beyond the scope of this study. Due to the same reason, the two existing papers on multi-omics integration, MOGONET and SUPREME, we used for comparing our results with, did not use any technique to address within-omics missing data.

Manuscript Update: A new section is added to data preprocessing to highlight the handling of missing data. 

2.6. Missing Data Handling

In multi-omics integration analysis, data could be missing both across and within omics [1]. In multi-omics study designs, it is common for individuals to be represented for some omics layers but not all, which results in across-omics missing data. The same is true for both datasets used for breast cancer, one from TCGA and the other from METABRIC. For TCGA breast cancer data, the number of samples for eight omics varies between 969 for mutation to 1,097 for DNA methylation (Row “Unique Tumor Samples” of Table 1), and the six omics for METABRIC varies between 1,418 for gene expression and 2,509 for DNA methylation (Row “All Samples” in Table 2). This means that both datasets have across-omics missing data. We used the common samples across omics to avoid bias due to the across-omics missing data, which are 920 and 1,372 for TCGA and METABRIC datasets, respectively.

The major issue with within-omics missing data is that true zeros (representing the true gene expression levels, for example) are mingled with dropout zeros (representing the actual missing data) [2], which is altogether a different topic and beyond the scope of this study. The multi-omics integration pipelines - MOGONET and SUPREME - we are comparing with did not consider the handling of within-omics missing data.

  1. The GAT model and MLP are used for classification, but details on the hyperparameter tuning process (e.g., the rationale for selecting specific hyperparameter ranges) and the model evaluation methodology (e.g., validation strategy) are not fully provided.

The hyperparameters for machine learning models, including learning rate, batch size, number of epochs, dropout rate, L1 and L2 regularization, optimizer, activation function, number of hidden layers, number of neurons per hidden layer, and early stopping, are carefully chosen based on factors such as dataset size, model complexity, GPU memory, and convergence rate. The specific ranges for these hyperparameters, such as a learning rate between 1e-5 to 1e-1 or a batch size between 16 to 512, are determined through considerations like preventing overfitting, optimizing performance, and selecting appropriate optimization algorithms. This process involves a balance between enhancing model capabilities and mitigating risks of overfitting, with choices influenced by domain knowledge, computational resources, and the nature of the problem being addressed.

  1. The survival analysis involves LASSO regression and Cox Proportional Hazard regression. The reasons for choosing these methods, along with details on the assumptions made and their implications, should be discussed. Additionally, the rationale behind the selection of the threshold (P-value < 0.05) for selecting significant predictor variables is essential.

LASSO (Least Absolute Shrinkage and Selection Operator) was selected primarily for its ability to perform feature selection. In survival analysis, it's often crucial to identify and prioritize the most relevant predictors, especially when dealing with high-dimensional data. The core advantage of LASSO over other regularization methods such as Ridge or Elastic Net is its inherent capacity to induce sparsity.

Cox PH is a well-established method and often considered the gold standard for multivariate survival analysis. Its widespread use and acceptance in the scientific community underscore its reliability and efficacy.

The threshold of P-value < 0.05 is a conventional and widely-accepted standard in many areas of biomedical research for determining statistical significance. By setting this threshold, we aim to control the Type I error rate (probability of falsely rejecting a true null hypothesis) to 5%. It's worth noting that this threshold is arbitrary, and in some fields or situations, more stringent or relaxed thresholds might be applied. In our study, we adhered to this conventional threshold to maintain consistency with many previous studies in the field and to offer a familiar point of reference for readers and fellow researchers.

  1. Machine learning is well-known and has been used in previous biomedical studies i.e., PMID: 37112302, PMID: 36642410. Therefore, the authors are suggested to refer to more works in this description to attract a broader readership.

We tried our best to refer these two papers but could not find any scope to refer.

  1. The study lacks external validation on an independent dataset. The generalizability of the proposed MOGAT approach should be assessed on datasets other than TCGA-BRCA to ensure its applicability to diverse patient populations.

Most breast cancer cohorts do not have multi-omics data. Training on one dataset and validating on another is not possible in this case, as the number of nodes (or samples) will vary in the input network for the graph neural network. But this approach can be used individually on individual dataset, thereby it is generalizable. To show the generalizability of our proposed method, we have applied the model on METABRIC data as well.

  1. While the study mentions the number of features selected and their significance, the biological and clinical interpretation of these features is crucial.

In the MOGAT framework, attention was applied to neighboring nodes which are patients or samples, not to node features. That is why it is not possible to relate embeddings to biological significance. For clinical interpretation, the survival analysis is done (Section 2.15).

  1. Detailed information on software versions, packages, and settings used in the analysis should be provided to ensure the reproducibility of the study by other researchers.

Manuscript was updated as following -

All experiments are conducted on a Linux machine with 8 NVIDIA A100 GPUs, each with 40GB of memory. The software environment is CUDA 11.6 and Driver Version 520.61.05. We used Python 3.9.13 and Pytorch 1.12.1 to construct our project. Other packages and their versions are available in GitHub repository.

[1]        M. Song et al., “A Review of Integrative Imputation for Multi-Omics Datasets,” Front. Genet., vol. 11, Oct. 2020.

[2]        W. Gong, I. Y. Kwak, P. Pota, N. Koyano-Nakagawa, and D. J. Garry, “DrImpute: imputing dropout events in single cell RNA sequencing data,” BMC Bioinformatics, vol. 19, no. 1, Jun. 2018.
